# Optimized superconductivity in the vicinity of a nematic quantum critical point in the kagome superconductor Cs(V₁₋ₓTiₓ)₃Sb₅

Yeahan Sur[1], Kwang-Tak Kim[1], Sukho Kim[1] & Kee Hoon Kim [1,2] ✉

$CsV_3Sb_5$ exhibits superconductivity at $T_c = 3.2$ K after undergoing intriguing two high-temperature transitions: charge density wave order at ~98 K and electronic nematic order at $T_{nem}$ ~ 35 K. Here, we investigate nematic susceptibility in single crystals of $Cs(V_{1-x}Ti_x)_3Sb_5$ ($x = 0.00$-$0.06$) where double-dome-shaped superconducting phase diagram is realized. The nematic susceptibility typically exhibits the Curie–Weiss behaviour above $T_{nem}$, which is monotonically decreased with $x$. Moreover, the Curie–Weiss temperature is systematically suppressed from ~30 K for $x = 0$ to ~4 K for $x = 0.0075$, resulting in a sign change at $x = $ ~0.009. Furthermore, the Curie constant reaches a maximum at $x = 0.01$, suggesting drastically enhanced nematic susceptibility near a putative nematic quantum critical point (NQCP) at $x = $ ~0.009. Strikingly, $T_c$ is enhanced up to ~4.1 K with full Meissner shielding realized at $x = $ ~0.0075-0.01, forming the first superconducting dome near the NQCP. Our findings directly point to a vital role of nematic fluctuations in enhancing the superconducting properties of $Cs(V_{1-x}Ti_x)_3Sb_5$.

Metals with kagome lattices have drawn enormous attention due to their unique electronic structures, which can host Dirac points, flat bands, and saddle points in their crystal momentum space. Many of these peculiar features can lead to electronic instabilities associated with divergence in the density of states near the Fermi level. As such, it has been theoretically predicted that various emergent orders such as charge bond order[1,2], charge density wave (CDW) order[1], nematic order[3], and superconductivity[1,2,4] might appear. For instance, a functional renormalisation group study on the kagome-Hubbard model predicted that CDW order and unconventional $d$-wave superconductivity can arise near a van Hove singularity at an electron filling of 5/4 per band[1].

Recently discovered V-based kagome metals $AV_3Sb_5$ ($A$ = K, Rb, and Cs) represent experimental realisations of these various theoretical predictions; each member indeed exhibits signatures of the structural transition due to the CDW order below its CDW transition temperature $T_{CDW}$, where $T_{CDW} = 80$-$104$ K[5–7], followed by the coexistence of superconductivity[8–10]. Within this material class, $CsV_3Sb_5$ has been most extensively studied due to its highest superconducting

transition temperature ($T_c$ = ~3.2 K) and the presence of multiple structural and electronic instabilities. For example, the CDW order found at $T_{CDW} = 98$ K[7] induces the first-order structural transition with a 2 × 2 modulation, showing a translational symmetry breaking in each kagome layer. Moreover, it was later found that the 2 × 2 modulation is accompanied by two kinds of out-of-plane modulations (i.e., 2 × 2 × 2[11] and 2 × 2 × 4[12]). The structure in the CDW states with the $c$-axis modulation is compatible with the $C_2$ rotational symmetry below $T_{CDW}$[13]. However, it is known that the electronic rotational symmetry in each kagome layer is still maintained as $C_6$ just below $T_{CDW}$[14]. With a further decrease in temperature, the system finally reaches the electronic nematic transition at $T_{nem}$ ~ 35 K, at which a rotational symmetry of the electronic state is continuously varied from $C_6$ to $C_2$ in the kagome layer, as evidenced by nuclear magnetic resonance (NMR), scanning tunnelling microscopy and elastoresistance measurements[14].

Alongside the experimental progresses, recent theoretical studies based on the kagome-Hubbard model have successfully explained the multiple thermal phase transitions related to the CDW and the nematic orders in $CsV_3Sb_5$[3,15–18]. Namely, the charge ordering at $T_{CDW}$ is

[1]Center for Novel States of Complex Materials Research, Department of Physics and Astronomy, Seoul National University, Seoul 08826, Republic of Korea. [2]Institute of Applied Physics, Department of Physics and Astronomy, Seoul National University, Seoul 08826, Republic of Korea. ✉e-mail: optopia@snu.ac.kr

understood as stemming from a triple-**q**-charge bond order, arising from the Fermi surface nesting at the van Hove singularity points in $CsV_3Sb_5$[3,15]. Moreover, near the suppression of these charge bond orders, it has been predicted that bond order fluctuations can mediate sizable pairing glue for superconductivity, possibly resulting in the superconductivity of various symmetries including singlet *s*-wave[17], triplet *p*-wave[17], or *d*-wave superconductivity[1].

However, in spite of extensive progress in both experiments and theories, the superconducting properties of $CsV_3Sb_5$ are still poorly understood. Firstly, no consensus has been reached on the pairing symmetry of $CsV_3Sb_5$. While unconventional superconductivity with the nodal gap structure has been suggested by thermal conductivity[19] and scanning tunnelling microscopy data[20], tunnel diode oscillator[21] and muon spin resonance measurements[22] support a nodeless *s*-wave gap with a sign-preserving order parameter. Secondly, an unusual superconducting phase diagram with a double-dome shape has been found independently either as a function of Sn doping in $CsV_3Sb_{5-x}Sn_x$ polycrystals[23] or that of pressure in a $CsV_3Sb_5$ single crystal[24,25], of which physical mechanisms remain elusive.

Elastoresistance measurements, which is a direct probe of the even-parity nematic susceptibility, have been found to be quite useful in unravelling the pivotal role of nematic fluctuation in mediating Cooper pairing, particularly in several Fe-based superconductors[26–30]. To better understand the role of nematic order in the superconductivity of $CsV_3Sb_5$, we employ elastoresistance measurements to systematically study the nematic susceptibility in high-quality single crystals of Ti-doped $CsV_3Sb_5$, $Cs(V_{1-x}Ti_x)_3Sb_5$. It should be emphasised that such studies have not been available thus far, as both controlling the doping ratios and maintaining high quality in the doped single crystals of $CsV_3Sb_5$ have been quite challenging[31,32].

## Results

Figure 1a depicts the crystal structure of $CsV_3Sb_5$, forming the hexagonal *P*6/*mmm* space group with lattice constants of $a = b = 5.508$ Å and $c = 9.326$ Å. One unit cell comprises a V-Sb layer sandwiched by Cs planes, each of which consists of four Cs atoms. Within each V-Sb sheet, a kagome network of V atoms is interlaced with a hexagonal lattice of Sb. As the temperature is lowered below $T_{CDW} = 98$ K, the V atoms rearrange themselves to form an inverse star-of-David pattern (2 × 2 charge order) in the kagome plane[7,33]. As the temperature is lowered further, an additional ordering is known to appear at $T_{nem}$ ~ 35 K, below which the system forms a distinct nematic phase[14] (Fig. 1b). Upon Ti doping, the titanium atoms can occupy the vanadium sites, resulting in progressive distortion of the vanadium kagome network.

Figure 1c shows the in-plane resistivity $\rho_{ab}$ of $Cs(V_{1-x}Ti_x)_3Sb_5$ normalised by the resistivity at 300 K ($\rho_{ab}/\rho_{ab,300 K}$). The resistivity of the undoped $CsV_3Sb_5$, evidencing metallic behaviour near room temperature, exhibits a slight shoulder near 98 K due to the formation of the CDW order. With increasing $x$, the residual resistivity ratio (RRR) of $Cs(V_{1-x}Ti_x)_3Sb_5$ systematically decreases. This result implies increased impurity scattering within the kagome plane in proportion to $x$. Along

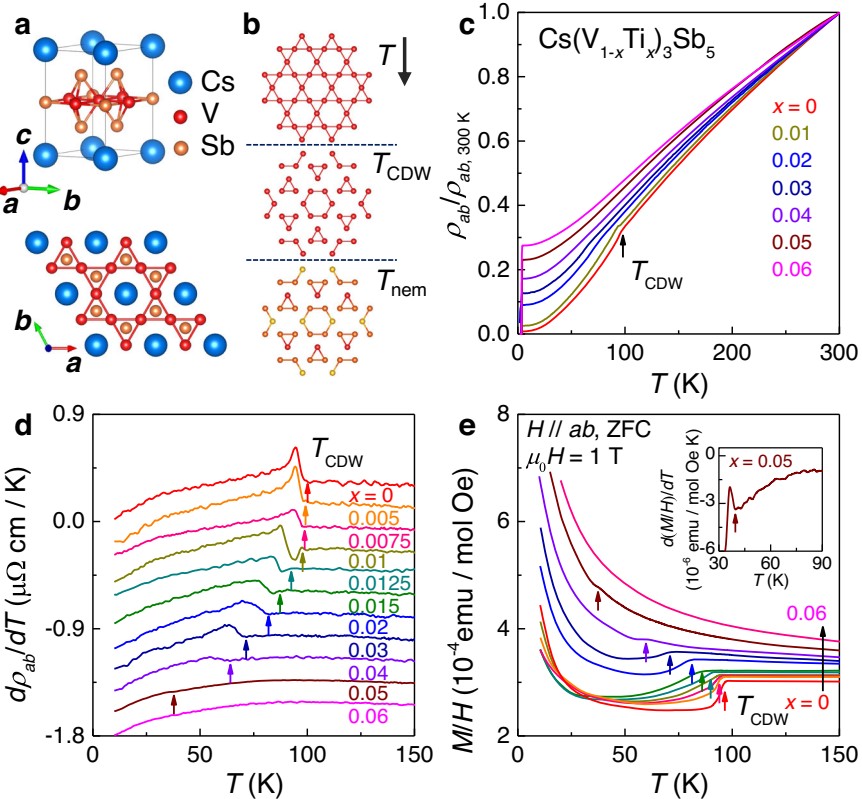

**Fig. 1 | Structure, in-plane resistivity ($\rho_{ab}$) and magnetic susceptibility ($M/H$) of $Cs(V_{1-x}Ti_x)_3Sb_5$.** **a** The crystal structure of $CsV_3Sb_5$. In the $V_3Sb_5$ layer, vanadium atoms form a kagome network within the $ab$ plane. **b** Schematic illustration of the temperature-dependent evolution of the vanadium kagome net across the known phase transitions of $CsV_3Sb_5$. Here, the $T_{CDW}$ indicates the charge density wave (CDW) transition temperature, while $T_{nem}$ indicates the nematic transition temperature. The red (orange) coloured vanadium atoms describe the maximum (minimum) of the local electron density of states in the nematic state, as observed from differential conductance maps from a recent scanning tunnelling microscopy

experiment[14]. **c** Temperature dependence of the in-plane resistivity ($\rho_{ab}$) normalised by the resistivity value of $Cs(V_{1-x}Ti_x)_3Sb_5$ single crystals at 300 K for $0 \leq x \leq 0.06$. **d** $d\rho_{ab}/dT$ curves of $Cs(V_{1-x}Ti_x)_3Sb_5$. The curves have been vertically shifted for clarity. The coloured arrows indicate $T_{CDW}$. **e** Temperature dependence of the $M/H$ of $Cs(V_{1-x}Ti_x)_3Sb_5$ measured at a constant external field of $\mu_0H = 1$ T after zero-field cooling (ZFC). Here, $M$ and $H$ indicate the magnetisation and magnetic field, respectively. The inset depicts the temperature-dependent $d(M/H)/dT$ plot of $Cs(V_{0.95}Ti_{0.05})_3Sb_5$ near $T_{CDW}$.

with the decreased RRR value, $Cs(V_{1-x}Ti_x)_3Sb_5$ exhibits increasingly broadened CDW transitions with increasing $x$, which are well visualised in the $d\rho_{ab}/dT$ curves (Fig. 1d). Furthermore, the anomalies in $d\rho_{ab}/dT$ shift to lower temperatures, indicating the development of a lower $T_{CDW}$ at higher $x$, e.g., ~59 K at $x = 0.04$. The decreasing trend of $T_{CDW}$ is also confirmed by the dc magnetic susceptibility ($M/H$) data presented in Fig. 1e. A drop in the $M/H$ curve observed in pristine $CsV_3Sb_5$, a Pauli paramagnet, indicates that depletion of the density of states has occurred due to the CDW gap opening at $T_{CDW} = 98$ K. Consistent with the behaviour found in $\rho_{ab}$, increasing $x$ results in a shift in $T_{CDW}$ (the temperature of the $M/H$ drop) to lower temperatures as well as a progressively decreased and broadened drop. This indicates that with progressive perturbation in the V atom arrangement by Ti doping, depletion of the electronic density of states at the long-range CDW transition is systematically reduced along with the decrease of the averaged $T_{CDW}$ and the increase of the $T_{CDW}$ distribution.

As the systematic suppression of $T_{CDW}$ is established with Ti doping, we now turn to the evolution of superconductivity to understand its interplay with the pre-existing CDW order. Figure 2 shows low-temperature $\rho_{ab}(T)$ and $4\pi\chi(T)$ data of $Cs(V_{1-x}Ti_x)_3Sb_5$. $T_c$ is clearly identified by either zero resistivity or the onset of diamagnetic behaviour in the $\chi(T)$ curves. Here, we define $T_c$ from the transport data by the criterion of $0.5\,\rho_N$ ($\rho_N$: normal-state resistivity)[34]. It is noted that $T_c$ exhibits nonmonotonic behaviour with $x$. At $x = 0.0075$, $T_c$ is maximised to 4.1 K from $T_c = 3.2$ K for the pristine compound. Moreover, in the doping range of $x = 0.0$–$0.01$, a full Meissner shielding of $-4\pi\chi \cong 1$ is realised. However, with a further increase of $x$ towards $x = 0.02$, $T_c$ is progressively suppressed to ~1.9 K, accompanied by decreased Meissner shielding of $-4\pi\chi$ ~ 0.3 at $x = 0.015$ and $-4\pi\chi$ ~ 0 around $x = 0.02$. At $x > 0.02$, $T_c$ increases again to 3.8 K at $x = 0.05$ and recovers the full Meissner shielding. Thus, the evolution of the superconducting properties, i.e., $T_c$ and Meissner shielding, reveals a double-dome feature.

Finding a superconducting region with a double-dome feature among CDW materials is unusual because the competition between CDW and superconductivity often leads to either a monotonically extended region of superconductivity after the collapse of the CDW order[35], or a single superconducting dome stabilised at the putative CDW quantum critical point (QCP)[36]. While an increase in $T_c$ at $x = $ ~0.05 may be associated with the putative QCP of the CDW order,

identifying another superconducting dome within the CDW order is truly uncommon. This finding strongly suggests that additional fluctuating orders could be present, enhancing the pairing interaction.

To investigate the origin of the unusual trend in $T_c$, the elastoresistance in $Cs(V_{1-x}Ti_x)_3Sb_5$ was investigated from 6 to 250 K. Figure 3a illustrates the experimental configuration for the elastoresistance measurements. It is known that nematic susceptibility $\tilde{n}$ can be obtained by measuring the electronic anisotropy induced by anisotropic strain. In other words, $\tilde{n}$ becomes linearly proportional to the anisotropic change in the resistance $N \equiv (\Delta R/R)_{xx} - (\Delta R/R)_{yy}$ in response to anisotropic strain $(\varepsilon_{xx} - \varepsilon_{yy})$, which results in $\tilde{n} = \alpha\partial N/\partial(\varepsilon_{xx} - \varepsilon_{yy})$. Here, $\alpha$ is a proportionality constant depending on microscopic details of the electronic structure[37] (See, Supplementary Note 8, for details). Therefore, $\partial N/\partial(\varepsilon_{xx} - \varepsilon_{yy})$ reveals the essential temperature dependence of $\tilde{n}$ upon approaching a thermally driven nematic phase transition. According to the general definition of elastoresistance coefficients $(\Delta R/R)_i \equiv \sum_{j=1}^{6} m_{ij}\varepsilon_j$[37], $\partial N/\partial(\varepsilon_{xx} - \varepsilon_{yy})$ can be expressed in terms of $m_{ij}$, where $\varepsilon_j$ represents the engineering strain, $m_{ij}$ are elastoresistance tensor components, and the subscripts $i$ and $j$ represent the Voigt notation ($1 = xx$, $2 = yy$, $3 = zz$, $4 = yz$, $5 = zx$, $6 = xy$). For a crystal in the $D_{6h}$ point group with $x$ measured along the [100] axis[14], $\partial N/\partial(\varepsilon_{xx} - \varepsilon_{yy})$ directly corresponds to the elastoresistance coefficient $(m_{11} - m_{12})$, representing the nematic susceptibility $\tilde{n}$ along the even-parity $E_{2g}$ symmetry channel (See, Supplementary Note 4).

Figure 3b depicts the response of $N$ to $(\varepsilon_{xx} - \varepsilon_{yy})$ for $CsV_3Sb_5$ at selected temperatures. $N$ shows a linear relationship with $(\varepsilon_{xx} - \varepsilon_{yy})$, enabling a precise measurement of $\tilde{n}$ by obtaining $N/(\varepsilon_{xx} - \varepsilon_{yy})$ in a small strain limit of $(\varepsilon_{xx} - \varepsilon_{yy}) \rightarrow 0$. This parameter is redefined as $\tilde{n} = \partial N/\partial(\varepsilon_{xx} - \varepsilon_{yy})$ ($\alpha$ is set to be 1), and the resulting $\tilde{n}(T)$ curve for $CsV_3Sb_5$ is presented in the top panel of Fig. 3c. A sharp jump in $\tilde{n}$ is found at $T_{CDW} = 98$ K, implying that the first-order structural transition[38] due to the charge bond order results in an abrupt offset change in the elastoresistance anisotropy. It is found that at temperatures above 36 K and below the sharp jump near $T_{CDW}$, $\tilde{n}$ is well fitted by the Curie–Weiss-type temperature dependence,

$$\tilde{n} = \tilde{n}_0 + \frac{C}{T - \theta_{nem}}. \tag{1}$$

Here, $\tilde{n}_0$ describes the intrinsic anisotropy in the piezoresistivity effect, unrelated to electronic nematicity, $\theta_{nem}$ is the mean-field nematic transition temperature, and $C$ is the Curie constant of the corresponding nematic susceptibility. A good agreement of the experimental data to Eq. (1) can be confirmed by a fitted red solid line with $\theta_{nem} = 30$ K, $\tilde{n}_0 = 14.7$, and $C = 39$ K. A good fit to Eq. (1) can also be verified by the plots of $(\tilde{n} - \tilde{n}_0)^{-1}$ and $(\tilde{n} - \tilde{n}_0)(T - \theta_{nem})$ at the bottom panel of Fig. 3c, which exhibit linear (pink open circles) and constant (green open circles) behaviours with temperature, respectively. The nearly constant value of $(\tilde{n} - \tilde{n}_0)(T - \theta_{nem})$ should directly correspond to the $C$ value.

It should be further noted in Fig. 3c that below 36 K as represented by a black dotted line, the experimental data clearly deviate from the trace predicted by Eq. (1). This deviation indicates that the nematic correlation goes beyond the mean-field description below this temperature located near the long-range ordering temperature $T_{nem}$. The temperature of 36 K is indeed close to $T_{nem}$ ~ 35 K, as found by the previous work[14]. Therefore, our observation shows that a significant nematic correlation of even-parity type exists above $T_{nem}$ up to $T_{CDW}$ and even above $T_{CDW}$ (vide infra).

Similar measurements and analyses were performed for $Cs(V_{1-x}Ti_x)_3Sb_5$ up to $x = 0.06$ (Fig. 3d–i) (for plots with $x \geq 0.03$, see Supplementary Fig. 6). It is noted that the jump in $\tilde{n}$ at $T_{CDW}$ systematically decreases with increasing $x$, indicating weakened elastoresistance anisotropy at the CDW ordering. More importantly, we find that

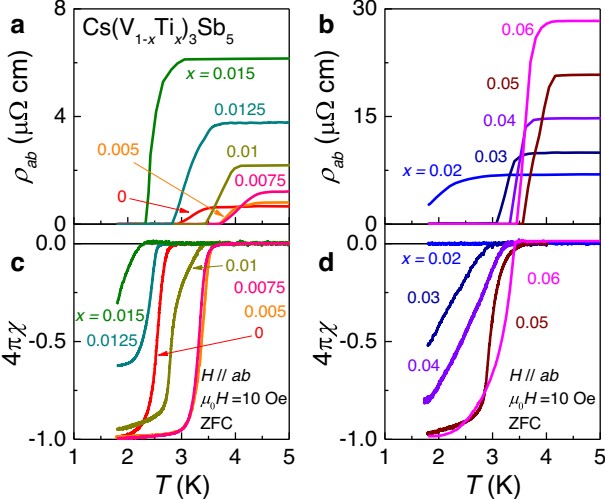

**Fig. 2 | Superconducting properties of $Cs(V_{1-x}Ti_x)_3Sb_5$.** Low-temperature behaviour of the in-plane resistivity $\rho_{ab}$ in $Cs(V_{1-x}Ti_x)_3Sb_5$ for **a** $0 \leq x \leq 0.015$ and **b** $0.02 \leq x \leq 0.06$. Temperature dependence of $4\pi\chi$ ($\chi$: magnetic susceptibility) measured at $\mu_0H = 10$ Oe after zero-field cooling (ZFC) for **c** $0 \leq x \leq 0.015$ and **d** $0.02 \leq x \leq 0.06$.

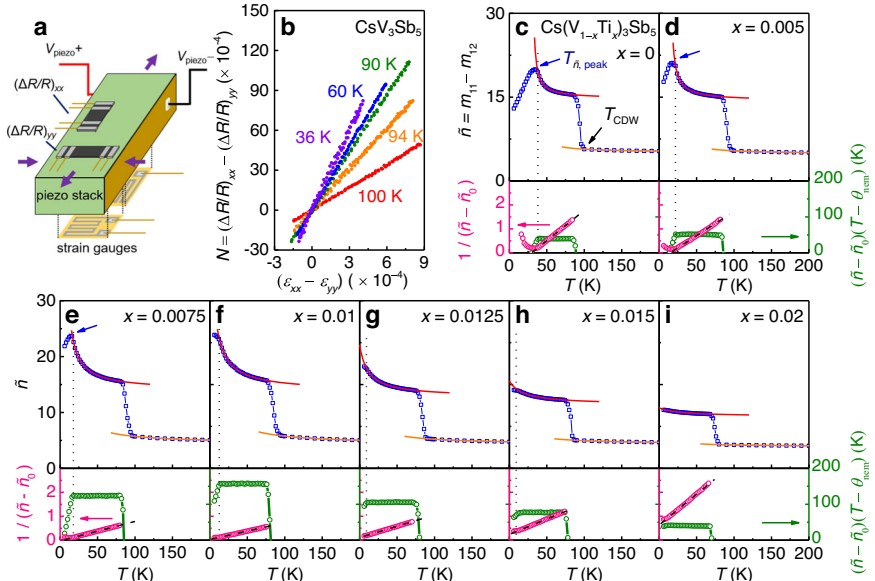

**Fig. 3 | Experimental methods for nematic susceptibility measurements and the temperature-dependent nematic susceptibility in Cs(V$_{1-x}$Ti$_x$)$_3$Sb$_5$.**
**a** Schematic illustration of the elastoresistance measurement setup. Here, $V_{piezo}$ indicates the voltage applied on the piezo stack. **b** $N$ vs. $(\varepsilon_{xx} - \varepsilon_{yy})$ plot of CsV$_3$Sb$_5$ at several representative temperatures, exhibiting a linear relationship; the nematic susceptibility $\tilde{n}$ can be obtained as $\partial N/\partial(\varepsilon_{xx} - \varepsilon_{yy})$. **c–i** (top panel) Temperature dependence of $\tilde{n}$ in Cs(V$_{1-x}$Ti$_x$)$_3$Sb$_5$ for $0 \leq x \leq 0.02$. A sharp jump in $\tilde{n}$ is observed near $T_{CDW}$. Below and above this jump, the data can be clearly fitted to the Curie–Weiss formula (Eq. (1)) (red and orange solid lines, respectively). The black dotted line indicates the deviation temperature from a Curie–Weiss fit, while the blue arrow indicates the peak temperature of $\tilde{n}$, $T_{\tilde{n}, peak}$. **c–i** (bottom panel) Temperature dependence of $(\tilde{n} - \tilde{n}_0)^{-1}$ and $(\tilde{n} - \tilde{n}_0)(T - \theta_{nem})$ below $T_{CDW}$, represented by the pink and green open circles, respectively. The black dashed line refers to a linear guideline.

all the samples up to $x = 0.03$ exhibit the Curie–Weiss temperature dependence of $\tilde{n}$ in broad temperature ranges above deviation temperatures represented by black dotted lines. For $x = 0$–$0.0075$, $\tilde{n}$ data clearly develop a peak at $T_{\tilde{n}, peak}$. In previous work on the pristine sample[14], $T_{nem}$ identified by NMR and $T_{\tilde{n}, peak}$ determined by the $\tilde{n}$ measurements were indeed nearly the same as ~35 K. Therefore, $T_{\tilde{n}, peak}$ can be used as a good estimate of $T_{nem}$ for each doping[39]. In our case, the resultant $T_{\tilde{n}, peak} = 34$ K for $x = 0$ is indeed close to the known value of $T_{nem} = $~35 K[14]. For other $x$, $T_{\tilde{n}, peak}$ shows a monotonous decrease; $T_{\tilde{n}, peak} = 18$ K ($x = 0.005$) and 14 K ($x = 0.0075$). For $x = 0.01$, however, we did not identify any peak feature at least down to 6 K, except finding the deviation temperature from the Curie–Weiss behaviour at ~12 K. This observation indicates that for $x = 0.01$, the true long-range nematic ordering is located well below 6 K or might not even exist at a finite temperature. For $x = 0.0125$ and 0.015, $\tilde{n}(T)$ doesn't show any peak feature either, and only the deviation from the Curie–Weiss behaviour is identified around ~8 K. This indicates that in $x = 0.0125$ and 0.015, only nematic correlation exists without the development of true long-range order at a finite temperature.

In order to understand quantitatively the evolution of $\tilde{n}$ over the broad doping ranges, we have tried to fit the experimental $\tilde{n}(T)$ of all the samples by Eq. (1) below $T_{CDW}$. (see Supplementary Table 2 for detailed fit parameters). Firstly, we discuss the evolution of $C$ for each doping. In contrast to the monotonic decrease in the jump of $\tilde{n}(T)$ and $T_{\tilde{n}, peak}$ with $x$, $C$ is found to exhibit nonmonotonic behaviour, i.e. increasing trend with $x$ for $x = 0.0$–$0.01$ and decreasing trend for $x \geq 0.0125$; $C$, as indicated by the slope of $(\tilde{n} - \tilde{n}_0)^{-1}$ or the value of $(\tilde{n} - \tilde{n}_0)(T - \theta_{nem})$, increases with $x$ from $C = 39$ K ($x = 0$) to 124 K ($x = 0.0075$), and exhibits a maximum value of $C = $~157 K at critical doping of $x_c = 0.01$. As a result, the highest value of $\tilde{n}(T = 6$ K) ~ 23.8 can be found at $x_c$. For $x \geq 0.0125$, $C$ decreases with $x$ to exhibit $C = 2$ K at $x = 0.03$, resulting in a flattening of the $\tilde{n}(T)$ curve at higher doping ratios. For $x \geq 0.04$, Eq. (1) cannot be fitted very well to the $\tilde{n}(T)$ curves due to the almost temperature-independent behaviour below and above the $T_{CDW}$.

The fit to Eq. (1) strikingly reveals that $\theta_{nem}$ is systematically suppressed from 30.0 K ($x = 0$) to 3.6 K ($x = 0.0075$), and to eventually exhibit a sign change ($x = $~0.009). At higher $x$, $\theta_{nem}$ is suppressed further, resulting in $\theta_{nem} = -42$ K at $x = 0.03$. In general, a nematic quantum critical point (NQCP) is often located at the phase space where $\theta_{nem}$ goes to zero temperature and strongly enhanced nematic susceptibility exists. The systematic suppression of $\theta_{nem}$ to zero temperature at $x = $~0.009, combined with the sharp maximum of the $C$ value and the disappearance of $T_{\tilde{n}, peak}$ near $x_c$, strongly suggests the presence of an NQCP near the doping level close to $x$ ~ 0.009-0.01. Indeed, similar phenomena have been observed in numerous Fe-based systems having the NQCP, such as Ba(Fe$_{1-x}$Co$_x$)$_2$As$_2$[27], Fe(Se$_{1-x}$S$_x$)[28], LaFe$_{1-x}$Co$_x$AsO[29], and Fe(Se$_{1-x}$Te$_x$)[30].

Another salient feature found in Fig. 3 is that the fit based on Eq. (1) below $T_{CDW}$ for each doping level can be successfully extended to explain $\tilde{n}(T)$ above $T_{CDW}$ with the same $\theta_{nem}$ and $C$ but with a different $\tilde{n}_0$ (orange solid line). This observation indicates that the even-parity nematic correlation might persist even above $T_{CDW}$ for CsV$_3$Sb$_5$ and Cs(V$_{1-x}$Ti$_x$)$_3$Sb$_5$ ($x \leq 0.03$). As a result, the Curie–Weiss tail above $T_{CDW}$ becomes clearly visible for $0 \leq x \leq 0.03$, reaching at least up to the maximum investigated temperature of 250 K (see, Supplementary Fig. 6 for $\tilde{n}(T)$ plots up to 250 K and for $x \geq 0.03$). Because of the most enhanced $C$ value, the Curie–Weiss tail of $\tilde{n}$ becomes most conspicuous at $x_c = 0.01$. It is striking that the enhanced $\tilde{n}$ up to at least 250 K is observed at $x_c = 0.01$ where the actual nematic ordering temperature is close to zero. This directly suggests that large quantum fluctuation of even-parity nematic order near the NQCP could be responsible for the enhancement of $\tilde{n}$ at high temperatures.

Figure 4a, b summarises the phase diagram of Cs(V$_{1-x}$Ti$_x$)$_3$Sb$_5$ plotted on top of the colour contour of $\tilde{n}$; $T_{CDW}$ as derived from the data of $\rho_{ab}$ (orange circles), $M/H$ (yellow octagons), and $\tilde{n}$ (brown crosses) are plotted for each $x$. Moreover, $T_{\tilde{n}, peak}$ (blue squares) and $\theta_{nem}$ (purple stars) obtained from $\tilde{n}$ in Fig. 3 are plotted with the $T_{nem}$ (pink cross) of $x = 0$ determined in a previous work[14]. At $x = 0$, a jump in $\tilde{n}(T)$ near $T_{CDW}$ can be clearly identified by the abrupt change of colour

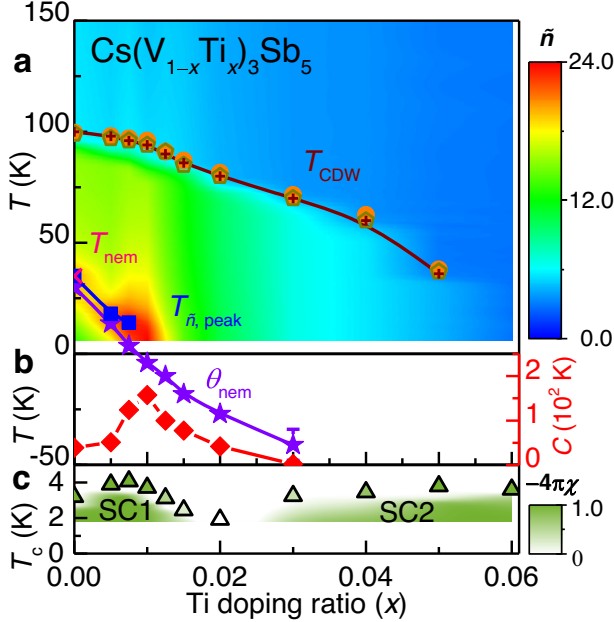

**Fig. 4 | Colour contour plot of nematic susceptibility, analyses results of the Curie–Weiss fit, and electronic transition temperatures of Cs(V$_{1-x}$Ti$_x$)$_3$Sb$_5$. a** The orange circles, yellow octagons, and brown crosses represent $T_{CDW}$ obtained from $\rho_{ab}$, $M/H$, and $\tilde{n}$ measurements. The pink cross in $T_{nem}$ indicates the nematic transition temperature obtained from ref. 14. The blue squares and purple stars represent $T_{\tilde{n},\,peak}$ and $\theta_{nem}$ obtained from the analysis of the nematic susceptibility $\tilde{n}$, respectively. A purple error bar is obtained by a standard deviation of the Curie–Weiss fit. The overlaid colour contour represents the interpolated values generated from $\tilde{n}$ taken from the elastoresistance measurements. **b** The red diamonds show the Curie constant $C$ obtained from the Curie–Weiss fits of $\tilde{n}$ for $0 \le x \le 0.03$. **c** Superconducting transition temperature $T_c$ (green triangles) defined by the criterion of $0.5\rho_N$ ($\rho_N$: normal-state resistivity) determined from Fig. 2. The colour contour represents the Meissner volume fraction ($-4\pi\chi$).

in $\tilde{n}$ from blue at $T > T_{CDW}$ to green at $T < T_{CDW}$. Near $T_{nem}$, the contour exhibits a yellow colour, indicating a local maximum of $\tilde{n}$ at $T_{\tilde{n},\,peak}$. With an increase in doping, the $T_{\tilde{n},\,peak}$ shifts to low temperatures, resulting in the most intensified $\tilde{n}$ (6 K) indicated by the red colour near $x = 0.0075$-0.01. This behaviour is also confirmed by the maximum of the $C$ value, indicated by the red diamonds.

The superconducting phase diagram of Cs(V$_{1-x}$Ti$_x$)$_3$Sb$_5$ is also plotted in Fig. 4c. Here, the green triangles indicate the $T_c$ obtained from transport measurements, while the Meissner volume fraction ($-4\pi\chi$) is represented below the trace of $T_c$ as a green colour contour. Surprisingly, it is found that $T_c$ is optimised to 4.1 K (3.7 K) near this doping range of $x = 0.0075$ (0.01), when the nematic correlations indicated by the $\tilde{n}$ and $C$ values are sharply enhanced near the NQCP. Our observation thus raises an intriguing possibility that fluctuation of the nematic order plays an important role in the pairing interaction to optimise superconductivity in the first superconducting dome of Cs(V$_{1-x}$Ti$_x$)$_3$Sb$_5$. At higher doping ratios of $0.01 \le x \le 0.02$, $T_c$ is monotonically suppressed with doping, which could be related to the reduced nematic fluctuations as indicated by the decreased $\tilde{n}$ and $C$ values.

Based on our experimental findings and implications, the nematic fluctuations may be important in understanding the superconductivity in the $A$V$_3$Sb$_5$ family. A very small $T_c \sim 0.0008$ K in CsV$_3$Sb$_5$ has been indeed predicted based on the McMillan equation[33], suggesting that electron-phonon coupling alone is not enough to explain the superconducting transition; nematic fluctuation should be considered as an essential ingredient to result in the relatively high $T_c \sim 3.2$ K in CsV$_3$Sb$_5$. In support of this scenario, it should be noted that the $T_c$ values in

KV$_3$Sb$_5$ ($T_c = 0.93$ K[8]) and in RbV$_3$Sb$_5$ ($T_c = 0.92$ K[9]) are lower than that in CsV$_3$Sb$_5$ ($T_c = 3.2$ K). In addition, unlike the Cs variant, a recent study of Sn doping in polycrystalline KV$_3$Sb$_5$ and RbV$_3$Sb$_5$ revealed single superconducting domes near the suppression of the CDW orders[40]. All these results, if interpreted correctly, potentially indicate that nematic order and its fluctuations might be absent in both compounds, motivating similar experiments for these materials. Furthermore, our scenario supports that nematic fluctuations should be also considered an important factor to understand the double superconducting domes reported in pressurised CsV$_3$Sb$_5$[24,25] and in CsV$_3$Sb$_{5-x}$Sn$_x$[23] polycrystals.

It should be pointed out that the experimental results found here well resemble those found in the Fe-based superconductors, e.g., Ba(Fe$_{1-x}$Co$_x$)$_2$As$_2$[27], LaFe$_{1-x}$Co$_x$AsO[29] and Fe(Se$_{1-x}$Te$_x$)[30], where unconventional superconductivity is optimised near the NQCP. However, the microscopic origin for having the nematic order seems to be quite different; in contrast to the iron-based superconductors where the spin density wave order is closely coupled to the nematic order[27–30,41], the nematic order in CsV$_3$Sb$_5$ is intertwined with the unconventional CDW order, possibly in a form of charge bond order[3,14,18]. Recent theoretical studies[3,18] considering the kagome-Hubbard model have indeed shown that a triple-**q**-charge bond order is stabilised below $T_{CDW}$, described by three complex CDW order parameters. Furthermore, those theories suggest that these CDW order parameters can undergo a continuous variation of their phases at $T_{nem}$ from a triply-degenerate, isotropic phase of $\pi/2$ into two different values, without the change of the isotropic amplitude, thereby resulting in one-dimensional nematic modulation. Therefore, if the charge bond order developed in CsV$_3$Sb$_5$ is assumed to be still maintained over the significant doping level $x$, the enhanced nematic correlation might be linked to the quantum phase transition involving a continuous variation of the phases of the triple-**q** CDW order parameters, at which the charge bond order with anisotropic phases, thus called nematic charge bond order, develops from the one with a homogenous phase.

Several other theoretical investigations considering the kagome-Hubbard model have also predicted that CsV$_3$Sb$_5$ might have unconventional superconductivity with a superconducting gap function of $s$- or $p$- or $d$-wave type when the CDW state is melted[1,17], possibly via suppression of the amplitudes in the three **q**-charge bond orders. Investigations on the superconducting order parameters at the doping level $x = -0.0075$-0.01 near the NQCP (first dome) and at $x = -0.05$ (second dome) may thus provide an opportunity for studying comparative characteristics of superconductivity instigated by anisotropic phase fluctuations and amplitudes of the three **q**-charge bond orders, respectively.

In conclusion, our experimental findings coherently suggest that an NQCP is located near $x = -0.009$-0.01. Moreover, a maximum $T_c = -4.1$ K with full Meissner shielding is realised at $x = -0.0075$-0.01, forming the first superconducting dome near the NQCP. This not only points out the vital role of nematic fluctuation in enhancing superconductivity but also provides important insights into understanding the link between the multiple orders and superconductivity in CsV$_3$Sb$_5$ and related kagome superconductors.

## Methods

### Single crystal growth

Cs(V$_{1-x}$Ti$_x$)$_3$Sb$_5$ ($0 \le x \le 0.06$) single crystals were grown by the Cs-Sb flux method. A mixture of Cs chunks (99.8%, Alfa Aesar), V powders (99.9%, Sigma Aldrich), Ti powders (99.99%, Alfa Aesar) and Sb shots (99.999%, Alfa Aesar) with a molar ratio of Cs: (V, Ti): Sb = 2: 1: 3 were put in an alumina crucible and were double-sealed in an evacuated quartz ampule with pressures less than $2 \times 10^{-5}$ mbar. The sealed ampules were heated at a rate of 100 °C/h and kept at 1000 °C for 24 h to fully dissolve the V and Ti into the Cs-Sb eutectic mixture. Later, the ampoules were cooled down to 600 °C at a rate of 2 °C/h. At 600 °C, the ampules were centrifuged to separate the crystals from the molten flux.

To avoid oxidation, all the sample preparation was done inside a glove box, which was kept in an Ar atmosphere with oxygen and moisture concentrations less than 1 ppm. Shiny plate-like crystals were obtained with a typical lateral area of $3 \times 2\,mm^2$. The $CsV_3Sb_5$ single crystals exhibited a residual resistivity ratio (RRR) value as high as 129.2.

### In-plane resistivity and magnetisation measurements

In-plane resistivity measurements were performed in the PPMS™ using a conventional four-probe method. The electrical contacts were attached by silver paint (Dupont 4929 N). The $M/H$ after zero-field cooling (ZFC) were obtained by MPMS™ (Quantum Design) at temperatures between 5 and 300 K, while the magnetic susceptibility $\chi$ between 1.8 and 5 K were measured by a vibrating sample magnetometer in a PPMS™ (Quantum Design). The demagnetisation factors in the $M/H$ and $\chi$ measurements have been corrected after measuring the samples in a needle-like configuration.

### Elastoresistance measurements

The elastoresistance was measured in a closed-cycle cryostat (Sumitomo RDK-101D) using a commercial piezoelectric actuator (Piezomechanik PSt 150) at various temperatures from 6 to 250 K. The samples were glued to the piezoelectric actuator by using an adhesive epoxy (Devcon 14250). Two samples were cut in rectangular shapes with a lateral size of ~$1 \times 0.2\,mm^2$; the longer direction was parallel (perpendicular) to the $a$-axis for the $R_{xx}$ ($R_{yy}$) sample. All the samples were cleaved to the thickness of ~20 μm to ensure efficient strain transmission to the entire sample. The electrical contacts were attached directly by silver paint (Dupont 4929 N) to the freshly cleaved surface. Two strain gauges were glued on the other side of the actuator with perpendicular orientation to each other.

### X-ray diffraction and wavelength dispersive X-ray spectroscopy measurements

$Cs(V_{1-x}Ti_x)_3Sb_5$ single crystals were ground and inserted inside a quartz capillary tube with an inner diameter of 0.5 mm. The tube was measured by x-ray diffraction (XRD) $\theta-2\theta$ scans using a high-resolution x-ray diffractometer (PANalytical Empyrean). Wavelength dispersive x-ray spectroscopy (WDS) were performed in a field emission electron probe microanalyzer (JEOL Ltd., JXA-8530F) by taking V (99.99%), Ti (99.9%) and Sb (99.99%) metals as standard specimens. The standard specimen data of Cs was taken from the JEOL database due to the high air sensitivity of the elemental Cs metal.

## Data availability

Source data are provided with this paper. The data generated in this study have been deposited in the Figshare database[42].

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

## Acknowledgements

Y.S., K.–T.K., S.K. and K.H.K. were financially supported by the Ministry of Science and ICT through the National Research Foundation of Korea (2019R1A2C2090648, 2022H1D3A3A01077468) and by the Ministry of Education (2021R1A6C101B418).

## Author contributions

Y.S. and K.H.K. initiated the project. Y.S., K.–T.K. and S.K. prepared the single crystalline samples. Y.S. characterised the samples and performed the measurements. Y.S. and K.H.K. analysed the data and wrote the manuscript. K.H.K. devised the project and advised the research. All authors discussed the results and commented on the manuscript.

## Competing interests

The authors declare no competing interests.
