## [Peer Review File · Nature Communications]

Optimized superconductivity in the vicinity of a nematic quantum critical point in the kagome superconductor Cs(V_{1-x}Ti_x)₃Sb₅Editorial Note: This manuscript has been previously reviewed at another journal that is not operating a transparent peer review scheme. This document only contains reviewer comments and rebuttal letters for versions considered at *Nature Communications*.

Reviewer #1 (Remarks to the Author):

The revised version is much improved. The authors have seriously considered and addressed the critics from me and fellow reviewers. The new measurements on extra samples and additional single-sample Montgomery method have validated the nematic critical point as well as the legitimacy of the two-sample method. The rewritten discussion has more depth on reasoning that connects the nematicity and the enhanced superconductivity which was my main concern in the first round of review. I recommend its publication on Nature Communications without further suggestions.

Reviewer #4 (Remarks to the Author):

The authors have addressed in detail all the points raised by referee #2. Their responses are thorough, convincing and adequate. I believe that the manuscript was also significantly improved in this process. Overall, the data is extremely clear and well presented. I would recommend publication of the article as is.

REVIEWERS' COMMENTS

Reviewer #1 (Remarks to the Author):

The revised version is much improved. The authors have seriously considered and addressed the critics from me and fellow reviewers. The new measurements on extra samples and additional single-sample Montgomery method have validated the nematic critical point as well as the legitimacy of the two-sample method. The rewritten discussion has more depth on reasoning that connects the nematicity and the enhanced superconductivity which was my main concern in the first round of review. I recommend its publication on Nature Communications without further suggestions.

>> We thank the reviewer #1 for acknowledging the significance of our research and recommending its publication in Nature Communications without any further suggestions.

Reviewer #4 (Remarks to the Author):

The authors have addressed in detail all the points raised by referee #2. Their responses are thorough, convincing and adequate. I believe that the manuscript was also significantly improved in this process. Overall, the data is extremely clear and well presented. I would recommend publication of the article as is.

>> We express our appreciation to Reviewer #4 for recognizing the high quality of our data and the clear presentation of our research. Furthermore, we are grateful to the reviewer for endorsing the publication of our article without any additional recommendations.